# The effects of sex and outcome expectancies on perceptions of sexual harassment

**Shonagh Leigh** *, **Andrew G. Thomas, Jason Davies**

Department of Psychology, Swansea University, Swansea, United Kingdom

* sholeigh84@gmail.com

**Data Availability Statement:** All relevant data are within the paper and its Supporting information files.

**Funding:** This work was funded collaboratively by the Welsh Government and KESS 2. Knowledge

## Abstract

Using an outcome expectancy framework, this research sought to understand sex differences in the underlying beliefs that influence harassment perception. One hundred and ninety-six participants (52% women) read a series of vignettes depicting common examples of digital male-on-female sexual harassment. They were asked to what extent they thought each scenario constituted sexual harassment, and how likely the perpetrator would experience positive and negative outcomes. Consistent with predictions, women were more likely to consider the behaviours as harassment than men were. Both sexes harassment perceptions had significant relationships with their outcome expectancies, but we also found evidence of a sex specific moderation; the link between men's negative outcome expectancies was moderated by their positive ones. The results suggest that perceptions of harassment may have sexually asymmetrical underpinnings. Measuring the interplay between positive and negative outcome expectancies in relation to sexual harassment perception is a novel approach, that may have implications for the development of anti-sexual harassment interventions. Implications for theory and future research directions are discussed.

## Introduction

Sexual harassment occurs in every known culture [1, 2]. Despite high prevalence and great public interest in reducing it, reviews of sexual harassment prevention strategies reveal a shortage of rigorous study [3–5]. Current interventions, such as workplace and online training, are frequently unempirical in development and assessment, with little focus on the perpetrator as an individual entity with personal goals [6, 7]. While there has been recent success understanding how and when sexual harassment is reported, interventions which reduce it are often ineffective or inconsistent [3, 4, 8]. This may be due, in part, to the minimal consideration given to the role of individual differences in harassment interventions. That is, training is not tailored toward those groups more likely to engage in sexual harassment, such as those who are young, sexist, or high in dark triad traits (narcissism, psychopathy, and Machiavellianism) despite clear associations between these factors and sexual harassment attitudes and engagement [9–11]. A targeted approach, developed with an appreciation of the unique underpinnings of sexual harassment within a particular sub-group, may lead to more effective reduction.

A common factor noted within legal reports of sexual harassment is sex; men are most likely to be perpetrators and women most likely to be victims [12, 13]. Furthermore, despite men being more likely to perpetrate sexual harassment, men are more likely than women to

Economy Skills Scholarships (KESS 2) is a pan-Wales higher level skills initiative led by Bangor University on behalf of the HE sector in Wales. It is part funded by the Welsh Government's European Social Fund (ESF) convergence programme for West Wales and the Valleys (ESF; https://ec.europa.eu/esf/home.jspcatId=45&langId=en). The European Union's Convergence programme was administered by the Welsh Government (grant code EGR 0817-100 / EGR9818-100, awarded to SL). The funders had no role in study design, data collection and analysis, decision to publish, or preparation of the manuscript. There was no additional external funding received for this study.

**Competing interests:** The authors have declared that no competing interests exist.

react negatively in response to current, standard, sexual harassment training [14, 15]. This highlights the importance of tailoring an intervention to the target audience. Using a "one size fits all" anti-harassment intervention strategy may be less effective because they assume that the underpinnings of harassment behaviour are identical between the sexes, despite well-documented sex differences in sexual psychology and behaviour [16–18]. Further, sexual harassment training invokes gender stereotypes; as typical sex-roles depict men as strong and overtly sexual and women as weak and conservative, men are typically perceived as perpetrators and women as victims within training sessions [19]. However, such interventions often fail to incorporate any psychological evidence-base for their structure [20]. If the sexes approach, interpret, and respond to sexual harassment in qualitatively different ways, then an appreciation for how the psychology of harassment differs by sex may allow for more customised and effective interventions.

Research and best-practice guidelines encourage the use of sex-specific programmes and advise careful programme formatting based on the target audience and desired outcomes [21, 22]. Unfortunately, there are many inconsistencies in how sexual harassment programmes are implemented and a lack of empirical findings being applied to programmes [20]. Current sex-specific interventions are underpinned by a message of "it's wrong, don't do it" despite this being a known trigger for resistance in training, leading to rebound effects [23]. Labelling individuals as problematic is likely to activate defensive behaviour as a method of cognitive self-preservation [23]. By assessing sex differences, traits, and cognitive processes surrounding sexual harassment, it may be possible to develop an approach that does not label individuals as potential perpetrators, but instead teaches realistic outcome expectancies and prosocial tactics to all, in a manner that appeals to the traits and motivations of high-risk individuals.

How harassment identification and interpretation differ across individuals may have implications for intervention development. For example, women are more likely than men to perceive certain behaviours at harassment [24, 25]. How a behaviour is valued (e.g., lawful versus unlawful, socially acceptable versus unacceptable) can influence associated outcomes expectancies [26, 27]. It is therefore important to first understand perceptions (and differences in perceptions) of potentially harassing behaviours. Conversely however, outcome expectancies have also been argued to affect how an individual values a behaviour [28, 29]. In either case, outcome expectancies have been found to be reliable predictors of behaviour across a wide variety of domains [30, 31].

The motivation underlying sexual harassment is a matter of much debate and, without sufficient understanding of said motivation, an effective, goal-orientated intervention cannot be developed. For example, some sociocultural arguments view sexual harassment as a method of seeking or maintaining power [32, 33]. Socialisation and cultural norms may also facilitate and even encourage engagement in sexual harassment [34, 35]. High gender-role stress and hegemonic masculinity may result in sexual harassment where traditional gender-norms are threatened [36]. Biological and evolutionary arguments propose that men have evolved a predisposition to aggressive sexual tactics to increase the likelihood of successful reproduction [37]. Evolutionary psychology extends this view, explaining that sexual harassment is an aggressive mate-gaining tactic, with cross-sex harassment being a form of mate-signalling while same-sex harassment degrades sexual competitors [38, 39]. Clearly sexual harassment is a multifaceted behaviour. However, what these theories have in common is that sexual harassment is classed as a goal-directed behaviour.

There is merit in considering what insights can be drawn from theories related to goal-directed behavioural decision-making in other domains. Outcome expectancies (OEs), or anticipated consequences, are an integral part of Bandura's social-cognitive theory and influence how cognitive-behavioural therapies are structured [40, 41]. For example, outcome

expectancy has been shown to be significantly related to pathological gambling, particularly in men [42], whilst OEs play a role with self-efficacy in relation to engaging in self-injurious behaviour [43]. Positive OEs have been widely studied in relation to health behaviour and are part of behaviour change treatment approaches [40, 44, 45]. For example, those that expect to enjoy smoking e-cigarettes are more likely to engage in this form of smoking [46] and those who believe that consuming alcohol will increase their confidence are more likely to drink [47]. Simultaneously having a lack of belief in ones' ability to resist alcohol can exacerbate this likelihood [43, 48]. Aggression studies have noted a significant sex difference in aggressive behaviour that is mediated by positive OEs, with men anticipating greater benefits [30]. In men, higher positive OEs are associated with intimate partner violence perpetration and with poor treatment responses [49]. Lower negative expectancies are associated with men's increased aggression and sexual coercion [30, 50].

A review of the literature on OEs in relation to substance abuse found them to be predictive of engagement and successful cessation and, importantly, to be modifiable within treatment [51]. Within health behaviour interventions, positive OE perceptions are challenged, and the benefits of alternative behaviours are discussed. However, these alternative behaviours must outweigh the benefits of the problem behaviour (e.g., by bringing greater and/or easier access to gains); expected outcomes must be desirable outcomes to reinforce behaviour [29]. Modifying OEs can be an important part of the motivational development stage of therapy in those that do not have the intention to change [52]. This may be especially relevant to those that harass without intent.

It has long been recognised that individuals that engage in sexual offenses have lower expectancies of negative outcomes [e.g., 53, 54]. Two studies cited OEs as part of trialled sexual assault prevention programmes [55, 56] although it should be noted that sexual harassment is not in itself mentioned within these studies. The first of these studies focused on rape prevention in college males. All components, including negative OEs, were significantly altered (in the desired directions) between pre- and post-intervention [56]. The second study, using a mixed-gender video-based sexual assault prevention programme, did not successfully alter OEs [55]. The latter is not a limitation of OEs as a potential intervention target, but a likely indication that the approach used within that study was not effective. Both studies discussed only raising awareness of negative OEs (not reducing positives) and neither recorded any follow-up data on actual engagement in sexual offenses. Trialling this further, including the modification of positive OEs and follow-up data, in the context of sexual harassment would be a useful next step.

Whilst the manipulation of outcome expectancies (OEs) has proven an effective treatment approach in the domains discussed, they have not been applied further in contemporary sexual harassment interventions. Key to these treatment approaches are the individuals' goals [44] and their beliefs in effective methods of obtaining them. A recent study [49] discusses this limitation and investigated how positive and negative OEs were related to intimate partner violence. Within their findings, positive and negative OEs were independent of each other. Positive expectancies were associated with greater perpetration as well as other factors that can result in poor treatment outcomes. Conversely, negative expectancies were associated with negative perceptions of intimate partner violence and greater motivation toward treatment. This exemplifies the importance of examining, and ultimately forming an intervention that addresses both factors.

Given the increasing role of online dating in relationship formation [57] and online harassment being increasingly problematic and difficult to regulate [58, 59], this domain is the focus herein. Multiple studies have demonstrated the high risks of sexual abuse and harassment associated with online activity such as the use of dating applications [60–63]. However, as

discussed within all these studies, there is a need for research to focus on aspects that may inform intervention development. The study being conducted here seeks to further examine sex differences and gain insight into beliefs regarding the goal-achieving efficacy of harassment behaviours and the interplay between these motivational and inhibitory factors. With these goals in mind, a novel vignette-based paradigm was developed, and its' psychometric value assessed. This paradigm was necessary because well-established measures, such as the Likelihood to Sexually Harass Scale [64], the Sexual Harassment Attitude Scale [65], and the Illinois Sexual Harassment Myth Scale [66], while reliable and accurate in their domains, do not readily lend themselves to measuring positive and negative outcomes of harassment behaviours as separate entities, and many of their examples do not translate to an online environment.

We would predict that women would perceive greater rates of harassment due to the greater threat of harassment that women typically face. Sexual harassment is a more predominant factor in women's lives (as is evident in the previously mentioned prevalence rates and typical manifestations). Furthermore, men typically have a physical advantage over women in a potentially threatening situation and women suffer greater consequences of reputation damage in relation to sexual encounters [67]. Therefore, sexual harassment is a more salient topic for women and one they are more likely to be vigilant in detecting. In contrast, men are more sexually competitive, and this is reflected by observable differences in men and women's short-term sexual psychology [68]. Thus, men are arguably more likely to accept and even condone persistent sexual advances in the pursuit of a mate, making them less likely than women to identify such behaviour as harassment.

We would also predict that perceptions of sexual harassment behaviour would be tied with outcome expectancy and that anticipated positive outcomes would be inversely related to anticipated negative outcomes. However, as the decision-making literature shows [30, 50, 69], there may also be some interplay between negative and positive outcome expectancies in relation to harassment perception. The possibility of a moderation effect between positive and negative OEs has been stipulated as an avenue of future research [49, 69], one that has been considered herein. There appears to be a lack of investigation into positive and negative outcome expectancies as independent factors relating to sexual harassment; specifically, how these interact with one another, whether either carries more weight, and, consequentially, how they affect attitudes and behaviours both separately and together.

As discussed, there are sex differences within outcome expectancies relating to aggressive behaviours [30]. Studies demonstrate men to be more sexually aggressive than women [70] and some define sexual harassment as an aggressive mate-seeking tactic [71, 72], thus sex differences surrounding outcome expectancies should also be investigated. Due to differences in the short-term sexual psychology of men and women, we might expect outcome expectancy to be more intimately tied to men's perception of harassment than to women's. Should this be the case, this would suggest sexual harassment intervention development would benefit from using sex-specific approaches. Thus, this study will address positive and negative OEs as separate components that may have differential relationships with perceptions of harassment across different individuals.

## Hypotheses

H1. Women will perceive significantly greater levels of harassment within vignettes than men will.

H2. Positive and negative outcome expectancies of sexual harassment behaviours will have an inverse relationship.

H3. Higher positive outcome expectancies will be associated with lower perceptions of harassment and higher negative outcome expectancies with higher perceptions of harassment.

H4. Perceptions of sexual harassment will have a significantly stronger relationship with men's outcome expectancies than with women's outcome expectancies.

In addition to our main hypotheses, we also investigated, on an exploratory basis, what individual differences (e.g., personality traits, sexist attitudes) might be uniquely associated with OEs. Numerous individual differences have been linked with sexual harassment attitudes and proclivity across studies. Sexism [9], low agreeableness and conscientiousness [73], dark triad traits [11], sadistic tendencies [74], rape myth acceptance [9], and previous harassment experience and engagement [36, 72] have all been linked with sexual harassment attitudes and/or proclivity. The dark triad and sadism also have specific links with online harassment [75, 76]. Complementing the measure of sexual harassment beliefs with individual differences helps us to understand both *whom* to target and *how* to target them.

## Materials and methods

This research received written approval from the Swansea University Department of Psychology ethics committee (reference 1395).

### Participants

Participants were recruited to take part in a study on "Individual differences in how online behaviours are interpreted" using opportunity sampling. The survey was advertised through multiple websites, posters, and face-to-face recruitment. The majority of participants were recruited via social media (e.g., Facebook, Twitter, and Reddit) although multiple websites were used (e.g., survey listings, general chat forums, and professional network-based sites). A priori power analysis with G*Power (power = .90. α = .05) indicated that a sample of 99 participants was required for a multiple regression with three predictors anticipating a medium effect size. Thus, we aimed to recruit 100 participants of each sex to allow us to run sex-specific models [77].

Of the 196 participants recruited, 51.53% indicated that their assigned biological sex was female. Five participants indicated that their gender identity was different from their sex. Repeating the reported analyses using gender rather than sex yielded qualitatively similar results. Differences are indicated using notes where applicable. The participants' mean age was 30.74 (*SD* = 13.79), and they were mostly white (88.4%), middle-class (42.9%), and heterosexual (70.9%). The sample contained a mixture of those in full-time employment (31%) and education (43.1%) and a mixture of those single (38.8%) or in a committed relationship (33.7%). A small number of participants indicated previous engagement in sexual harassment (15.8% of men, 13.9% of women) and a greater number indicated previous personal experiences of sexual harassment (54.7% men, 69.3% women); of these participants, 12.6% of men and 10.9% of women had both experienced and engaged in sexual harassment. Participants completed the study either in exchange for participation credit (*n* = 23), or for no compensation.

### Materials

**Online and Digital Sexual Harassment Attitude Measure (OD-SHAM).** The OD-SHAM was developed to measure participant's perceptions of sexual harassment. It contained a series of 21 vignettes. Vignettes depicted typical examples of male-on-female sexual harassment in online and digital contexts. For example, "Via a dating website, after talking with Jane for some time, James sends an explicitly detailed sexual message". A range of online

and digital harassment behaviours were included, such as contacting friends and family for information about the target, monitoring a partner's social media, and disrupting the targets' relationship with her current partner. For each vignette, participants are asked how likely Jane is to consider James' behaviour harassment, how likely the behaviour would lead to a positive outcome (e.g., Jane agrees to meet with James), and how likely it would lead to a negative outcome (e.g., Jane reports James). All questions were responded to using a Likert scale (1 – *Extremely unlikely* to 7 –*Extremely Likely*).

There were originally 26 vignettes. Of these, five were excluded from final analyses as they were intentionally non/low harassment (e.g., "Via a dating website, James introduces himself to Jane with a message indicating his interest in Jane"), used to determine normative ratings. Positive and negative outcome expectancy scores across the vignettes showed good reliability (α = .79 and .82, respectively) and so these were summed into total scores reflecting the participants general cross-vignette positive outcome expectancies (POE) and negative outcome expectancies (NOE). Cross-vignette judgements of harassment also showed good reliability (α = .94) and so were summed to produce a global measure of harassment sensitivity (H). Similar reliability (H:α = .94; POE: α = .92; and NOE:α = .90) and no order effects were found in a small validation study (*n* = 38) conducted using the OD-SHAM alone for the sole purpose of confirming reliability.

**Measures of attitude and personality.**   As part of an exploratory analysis, participants completed measures of personality and attitude toward the opposite sex to examine the correlates of POE and NOE. Specifically, we measured personality (Big Five Inventory, α range = .78-.98; [78]), Machiavellianism, narcissism, and psychopathy (Short Dark Triad, α = .78; [79]), everyday sadistic tendencies (Short Sadistic Impulse Scale, α = .77; [80]), hostile and benevolent sexism (Ambivalent Sexism Inventory, α = .65; [81]), rape myth acceptance (RMA; Illinois Rape Myth Acceptance-Short Form, α = .86; [82]), and previous harassment experience and engagement (Harassment Behaviour Scale, α = .93 and .95 respectively; adapted from Turmanis & Brown; [83]). Alphas relate to the sample gathered in the present study.

## Procedure

The study began by participants providing informed consent and demographic information. The attitude and personality measures were then completed in the order presented above. Participants then completed the OD-SHAM with vignettes presented in a random order before receiving a full debrief. Ethical approval was granted by the ethics committee of [REDACTED].

## Results

Each vignette was examined and compared between the sexes to determine any particular behaviours of interest. Table 1 displays the means and SDs of vignettes H, POE, and NOE scores by sex and the effect size. Overall, participants felt that the actions from the vignettes were likely to be considered harassment by the receiver and lead to negative, rather than positive, responses (Table 1). Average scores for each sub-set of vignettes (five sub-sets in total) are also presented in Table 1.

Significant sex differences in perceptions of harassment are present within the relationship disruption sub-set. Men rated disruption items 1 and 2 as significantly more likely to have a positive outcome. The non-harassing approach behaviours sub-set which was not included in final harassment or outcome expectancy scores, also revealed a sex difference. Women rated item 1, "sending a hello message" as significantly more likely to have a positive outcome.

**Table 1. Summary statistics for the individual vignettes of the Online and Digital Sexual Harassment Measure (OD-SHAM) for men and women are presented separately.**

| | Initiating Contact | | | | | | | | |
|---|---|---|---|---|---|---|---|---|---|
| **Behaviour** | **Harassment Perception** | | | **Positive Outcome Expectancy** | | | **Negative Outcome Expectancy** | | |
| | Men | Women | d | Men | Women | d | Men | Women | d |
| 1. Hello message[1] | 1.26 (0.70) | 1.52 (.86) | .33 | 4.53 (1.54) | 5.34 (1.24) | **.58** | 2.82 (1.55) | 2.29 (1.42) | -.36 |
| 2. Well-dressed photo[1] | 1.47 (1.20) | 1.51 (.81) | .04 | 4.68 (1.42) | 5.13 (1.22) | .34 | 2.75 (1.46) | 2.43 (1.47) | -.22 |
| 3. Provocative photo[1] | 2.89 (1.38) | 3.27 (1.56) | .25 | 3.31 (1.23) | 3.75 (1.37) | .34 | 4.26 (1.20) | 3.90 (1.40) | -.24 |
| 4. Sexually explicit message | 5.73 (1.17) | 6.05 (1.17) | .28 | 1.89 (1.05) | 2.00 (1.24) | .09 | 5.98 (1.08) | 5.98 (1.14) | .00 |
| 5. Sexually explicit photograph | 6.52 (1.15) | 6.70 (.70) | .20 | 1.40 (.72) | 1.42 (1.05) | .02 | 6.57 (.81) | 6.73 (0.83) | .20 |
| **Sub-set average** (items 4 and 5) | 6.12 (0.98) | 6.38 (0.09) | .28 | 1.65 (0.81) | 1.71 (1.02) | .07 | 6.27 (0.83) | 6.36 (0.89) | .09 |
| | Relationship Pursuit | | | | | | | | |
| 1. Contacts friends/family online[1] | 3.67 (2.04) | 3.60 (1.96) | -.04 | 2.06 (1.20) | 2.02 (1.23) | -.04 | 5.60 (1.35) | 5.80 (1.48) | .14 |
| 2. Sexually suggestive message[1] | 3.68 (1.73) | 3.96 (1.71) | .16 | 3.26 (1.40) | 3.56 (1.43) | .21 | 4.57 (1.41) | 4.42 (1.43) | -.11 |
| 3. Sexually explicit message | 4.94 (1.51) | 5.23 (1.64) | .18 | 2.47 (1.24) | 2.65 (1.25) | .14 | 5.25 (1.40) | 5.25 (1.41) | .00 |
| 4. Sexually explicit photograph | 5.74 (1.41) | 5.80 (1.54) | .04 | 2.14 (1.29) | 2.18 (1.23) | .03 | 5.80 (1.36) | 5.78 (1.34) | -.01 |
| **Sub-set average** (items 3 and 4) | 5.34 (1.39) | 5.51 (1.48) | .12 | 2.31 (1.21) | 2.42 (1.17) | .09 | 5.52 (1.28) | 5.52 (1.28) | -.01 |
| | Retaliation to Rejection | | | | | | | | |
| 1. Daily messages | 4.85 (1.95) | 5.40 (1.48) | .32 | 1.80 (0.98) | 1.84 (1.20) | .04 | 6.09 (1.11) | 6.13 (1.21) | .04 |
| 2. Insults | 4.88 (2.06) | 5.16 (1.74) | .14 | 1.15 (0.44) | 1.15 (0.58) | -.00 | 6.63 (0.70) | 6.74 (0.66) | .16 |
| 3. Fake profile to try again | 5.49 (1.95) | 5.97 (1.51) | .27 | 1.22 (0.57) | 1.07 (0.26) | -.33 | 6.74 (0.53) | 6.84 (0.49) | .19 |
| 4. Contacts friends/family online | 5.20 (2.09) | 5.45 (1.86) | .12 | 1.26 (0.66) | 1.22 (0.63) | -.06 | 6.60 (0.78) | 6.66 (0.77) | .08 |
| 5. Threaten to self-harm | 4.58 (2.25) | 4.92 (2.19) | .15 | 1.27 (0.78) | 1.45 (1.21) | .18 | 6.53 (1.04) | 6.36 (1.34) | -.14 |
| 6. Revenge fake profile of Jane | 5.79 (1.79) | 6.02 (1.61) | .14 | 1.03 (0.18) | 1.11 (0.58) | .18 | 6.91 (0.41) | 6.76 (0.92) | -.21 |
| 7. Following offline | 6.22 (1.44) | 6.43 (1.16) | .16 | 1.14 (0.46) | 1.14 (0.60) | .00 | 6.75 (0.78) | 6.81 (0.78) | .07 |
| 8. Threaten to hurt Janes' friends/family | 5.86 (1.84) | 5.87 (1.90) | .00 | 1.03 (0.23) | 1.11 (0.70) | .15 | 6.86 (0.82) | 6.83 (0.90) | -.04 |
| **Sub-set average** | 5.36 (1.59) | 5.65 (1.34) | .19 | 1.24 (0.33) | 1.26 (0.45) | .06 | 6.64 (0.45) | 6.64 (0.52) | .00 |
| | Relationship Maintenance | | | | | | | | |
| 1. Tracking Jane online and via GPS | 4.34 (1.87) | 4.86 (1.88) | .27 | 1.52 (.91) | 1.76 (1.07) | .24 | 6.18 (1.14) | 6.10 (1.22) | -.07 |
| 2. Tracking Jane via her contacts | 4.35 (2.07) | 4.85 (1.95) | .25 | 1.66 (1.10) | 1.70 (1.09) | .04 | 6.08 (1.31) | 6.13 (1.26) | .04 |
| 3. Checking phone/computer | 4.36 (2.11) | 4.83 (1.98) | .23 | 1.31 (.74) | 1.40 (0.92) | .11 | 6.41 (.98) | 6.58 (0.76) | .20 |
| 4. Monitoring all digital interactions | 5.01 (2.17) | 5.25 (1.92) | .12 | 1.27 (0.92) | 1.24 (0.75) | -.03 | 6.67 (0.71) | 6.66 (0.93) | -.01 |
| 5. Loyalty test with fake online profile | 4.97 (2.08) | 5.26 (1.98) | .14 | 1.38 (1.08) | 1.16 (0.44) | -.27 | 6.66 (0.85) | 6.79 (0.52) | .19 |
| **Sub-set average** | 4.60 (1.87) | 5.01 (1.66) | .23 | 1.43 (0.67) | 1.45 (0.60) | .04 | 6.40 (0.78) | 6.45 (0.70) | .07 |
| | Relationship Disruption | | | | | | | | |
| 1. Stranger claims Jane is unfaithful | 4.53 (1.94) | 4.37 (1.94) | -.08 | 1.67 (1.23) | 1.14 (0.57) | **-.56** | 6.32 (1.12) | 6.67 (1.06) | .32 |
| 2. Friend claims Jane is unfaithful with himself | 4.86 (1.95) | 4.45 (1.95) | -.21 | 1.70 (1.37) | 1.21 (0.70) | **-.45** | 6.40 (1.12) | 6.70 (0.87) | .30 |
| 3. Ex-partner sends embarrassing information to Janes' partner | 5.84 (1.38) | 5.93 (1.48) | .06 | 1.48 (1.20) | 1.15 (0.70) | -.35 | 6.62 (1.04) | 6.79 (0.80) | .18 |
| 4. Ex-partner sends indecent images of Jane to her partner | 6.66 (0.75) | 6.56 (1.06) | -.11 | 1.23 (.90) | 1.16 (0.79) | -.08 | 6.83 (0.75) | 6.81 (0.90) | -.02 |
| **Sub-set average** | 5.47 (1.31) | 5.33 (1.31) | -.12 | 1.52 (1.01) | 1.17 (0.56) | **-.44** | 6.54 (0.84) | 6.74 (0.73) | .25 |
| [1] **Normality vignettes** | 2.60 (2.96) | 2.77 (2.75) | .25 | 3.57 (0.92) | 3.96 (0.76) | **.47** | 4.00 (0.93) | 3.77 (0.80) | -.27 |

| **Aggregate H** | | | **Aggregate POE** | | | **Aggregate NOE** | | |
|---|---|---|---|---|---|---|---|---|
| Men | Women | d | Men | Women | d | Men | Women | d |
| 5.19 (1.26) | 5.56 (1.05) | **.32** | 1.48 (0.41) | 1.44 (0.38) | -.09 | 6.42 (0.47) | 6.48 (0.45) | .13 |

[1] = normality vignettes, not included in final analysis or sub-set aggregate scores. d = Cohen's d effect size.

Effect sizes in bold are significant to the $p < .05$ level following Bonferroni correction.

**Table 2. Harassment perception and outcome expectancy correlations.**

| | H & POE | | H & NOE | | POE & NOE | |
|---|---|---|---|---|---|---|
| | *r* | *p* | *r* | *p* | *r* | *p* |
| **All** | -.33 | < .001 | .37 | < .001 | -.78 | < .001 |
| **Men** | -.41 | < .001 | .47 | < .001 | -.75 | < .001 |
| **Women** | -.23[1] | .02 | .25 | .01 | -.81 | < .001 |

H = harassment perception, POE = positive outcome expectancy, NOE = negative outcome expectancy.

[1] this correlation becomes nonsignificant when gender identity is used rather than assigned biological sex.

There was a sex difference in overall harassment perception; globally, women felt that James' actions were more likely to be perceived as harassment by Jane than men did. This difference was small-to-medium in size. Overall, men and women predicted that the actions would lead to positive and negative outcomes in similar ways.

Correlations between H, POE, and NOE scores revealed that harassment perception was positively associated with predicted negative outcomes, and negatively associated with positive outcomes (Table 2). These relationships were almost twice as strong in men than women ($z = 1.76$, $p = .04$ for H and NOE and $z = 1.39$, $p = .08$ for H and POE, one-tailed). Both sexes showed a very strong negative correlation between perceived positive and negative outcomes indicating that as those who felt that an action would lead to a positive outcome felt that negative outcomes were less likely. It is worth noting that the amount of variance shared between these two variables (61%) reflects the fact that some participants are ambivalent. Rather than being two sides of the same coin, some participants feel that actions may lead to positive as well as negative outcomes.

To examine the interplay between positive and negative outcome expectancies on perceptions of harassment, we conducted a hierarchical regression for both sexes (Table 3). We ran separate models for men and women because sex-specific analyses permit for possible sexually distinct and within-sex patterns to be observed [77]. This method of observation aligns with the goals of the current research. Step 1 contained POE and NOE while Step 2 added their means-centred interaction. The interaction between POE and NOE was found to be a significant predictor of men's harassment perception. No OEs were significant predictors of women's harassment perceptions.

**Table 3. Sex-specific models predicting harassment perception using positive outcome expectancies, negative outcome expectancies, and their interaction.**

| | Men | | | Women | | |
|---|---|---|---|---|---|---|
| | *B* | *SE* | *p* | *B* | *SE* | *p* |
| **Step 1** | | | | | | |
| POE | -0.12 | 0.43 | .39 | -0.09 | 0.46 | .60 |
| NOE | 0.38 | 0.38 | .01 | 0.18 | 0.39 | .30 |
| Model: | $F(2,90) = 13.43$, $p < .001$, $R^2 = .23$, Adj.$R^2 = .21$ | | | $F(2,97) = 3.32$, $p = .04$, $R^2 = .06$, Adj. $R^2 = .05$ | | |
| **Step 2** | | | | | | |
| POE | -0.29 | 0.43 | .05 | -0.14 | 0.48 | .42 |
| NOE | 0.39 | 0.37 | .01 | 0.22 | 0.40 | .20 |
| POE*NOE | -0.30 | 0.03 | .01 | -0.16 | 0.03 | .19 |
| Model: | $F(3,89) = 12.29$, $p < .001$, $R^2 = .29$, Adj.$R^2 = .27$ | | | $F(3,96) = 2.80$, $p = .04$, $R^2 = .08$, Adj.$R^2 = .05$ | | |

POE = Positive Outcome Expectancies, NOE = Negative Outcome Expectancies.

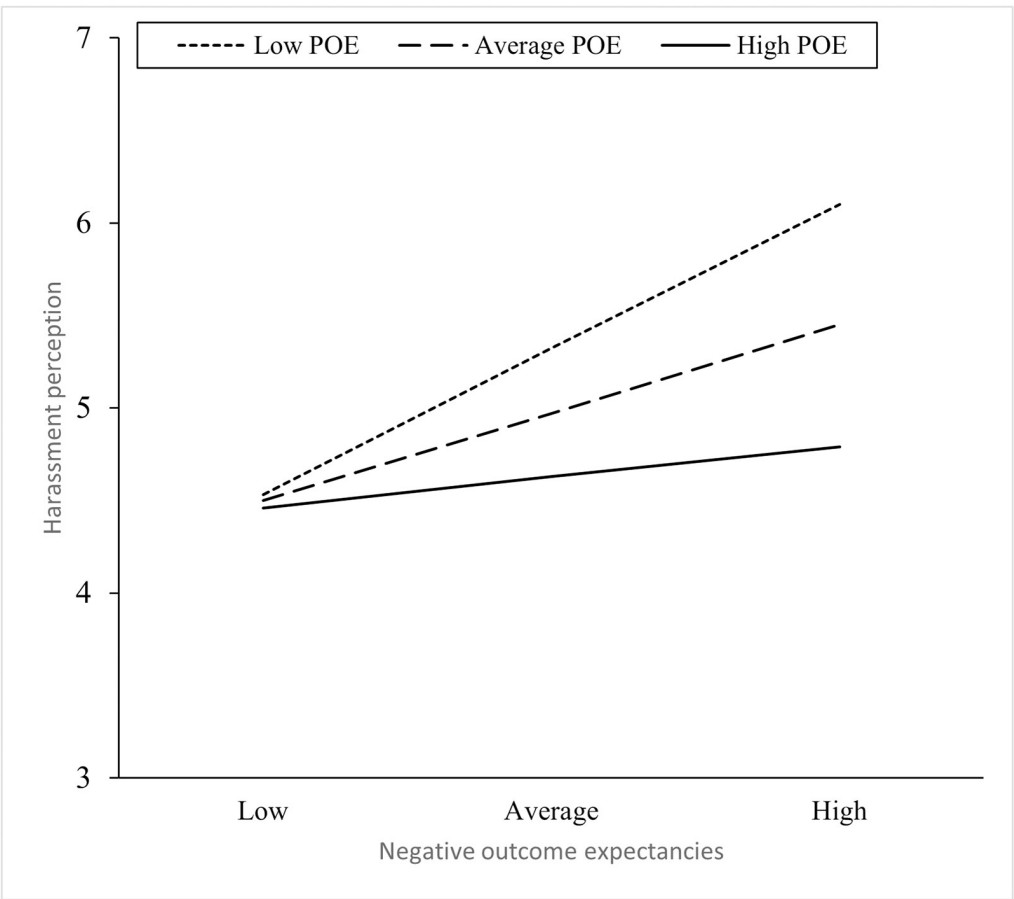

**Fig 1. Simple slopes plot displaying the moderation effect of positive outcome expectancies on the relationship between negative outcome expectancies and harassment perception.** POE = positive outcome expectancies.

Approximately a third of the sample identified as non-heterosexual. We re-analysed the data while excluding participants with same-sex preferences. This made no significant change to the results.

The resulting model in men accounted for over a quarter of the variance in H (27%). To better understand the interaction, moderation analysis was performed using the Hayes PROCESS macro [84]. This revealed a significant relationship between NOE and H when POE scores are low ($-1$ $SD$; $t(89) = 3.88$, $p < .001$) or average ($t(89) = 2.77$, $p = .01$) but not when they are high ($+1$ $SD$; $t(89) = .80$, $p = .56$). Thus, men who think that an action is likely to lead to a negative outcome are less inclined to view this action as harassment if they also feel the behaviour is likely to lead to a positive outcome (Fig 1).

For participants low ($-1$ $SD$) in PEO, the difference in harassment perception between those with low NEO and high NEO was 1.57. This represented an increase of more than one standard deviation in harassment perception, shifting response from the middle (4.53) of the scale toward the upper end (6.10). In contrast, for participants high ($+1$ $SD$) in PEO, the difference in harassment perception was 0.33, a negligible increase. The change in men's harassment perception (H) is displayed in Fig 1.

On an exploratory basis and to better understand factors that may predict a key moderator of sexual harassment perceptions, partial correlations were performed to examine what

**Table 4. Partial correlations of individual differences with positive outcome expectancies when controlling for negative outcome expectancies.**

| Variable | Men | Women |
|---|---|---|
| Rape Myth Acceptance | $r = .26$ | $r = .40$ |
| Hostile Sexism | $r = .28$ | $r = .17$ |
| Benevolent Sexism | $r = .22$ | $r = .07$ |
| Age | $r = .05$ | $r = .39$ |
| Sexual Harassment Experience | $r = -.03$ | $r = -.23$ |
| Agreeableness | $r = -.04$ | $r = .30$ |
| Conscientiousness | $r = -.03$ | $r = .03$ |
| Extraversion | $r = -.19$ | $r = .05$ |
| Neuroticism | $r = .02$ | $r = .06$ |
| Openness | $r = -.16$ | $r = -.11$ |
| Intrasexual Competitiveness | $r = -.03$ | $r = .23$[1] |
| Sadistic Tendencies | $r < -.01$ | $r = .33$ |

Correlations in bold are significant to the $p < .05$ level following Bonferroni correction.

[1] this correlation becomes nonsignificant when gender identity is used rather than assigned biological sex.

characteristics are associated with positive appraisals of sexual harassment when controlling for negative ones (Table 4). This is reflective of previous research in which positive and negative OEs have different predictors [49]. For correlations between all variables, see the (S1 Table). The results revealed that, for men, none of the individual differences measured herein uniquely correlated with positive outcome expectancies. For women, RMA and holding sadistic tendencies correlated with positive outcome expectancies and younger women were less likely to see positive outcomes from the behaviour.

## Discussion

This study examined sex differences in how positive and negative outcome expectancies (OEs) affected judgements of social interactions as sexual harassment. As predicted, and in accord with previous research [85], women were more likely to perceive the actions depicted in the vignettes as sexual harassment (they had higher H-scores) than men (H1). This was a small effect size that was detectable at the composite level ($d = 0.32$). Second, positive and negative OEs (POE and NOE scores) had a significant inverse relationship (H2). This relationship was strong, though 39% of the variance was unshared, suggesting some level of independence. This is fitting with recent findings in intimate partner violence research [49]. Third, POE and NOE scores correlated at similar levels with H-scores, though the strength of these correlations was nearly double for men, a sex difference that was significant for NOE and approached significance for POE. That OEs would predict H-scores (H3) was only partially supported; a significant model emerged for men but not women. These findings echo research in which sex differences in aggression were partially moderated by positive and negative OEs [30]. OEs and perceptions of harassment may not relate in the same manner for both sexes, having a stronger overall relationship for men (as predicted, H4). Finally, exploratory work revealed unique patterns associated with POE in women, adding weight to the idea that POE and NOE are distinct entities [49]. However, confounding this, no unique relationships were revealed for men's POE. There may be other individual differences that uniquely relate to men's POE that have not been examined here.

When investigating the unique contribution of POE and NOE to perceptions of harassment, we found an interaction between POE and NOE in the case of men which we pursued, on an exploratory basis, with moderation analysis. This analysis revealed that POE moderated the relationship between NOE with men's perceptions of sexual harassment. When POE was low or average, actions deemed likely to produce negative outcomes were linked to higher perceptions of harassment. This association disappeared when POE was high. This is highly similar to predictions in previous research and, as suggested by the authors, supports the notion of assessing whether differing combinations of positive and negative OEs are associated with specific characteristics and treatment needs [49].

We investigated variance in POE by considering its association with some traits known to predict harassment behaviour [9–11]. POE was uniquely related to rape myth acceptance, age, and sadistic tendencies in women, suggesting women high in these traits may be more likely to anticipate positive outcomes following harassment behaviours. In this study, for practicality, the focus was on traits with a long-standing evidence base of their association with sexual harassment [9, 71, 86]. It would be beneficial to examine other traits that may relate to men's POE, such as sociosexuality and attachment style, which have also demonstrated relationships with sexual harassment [61, 63]. Understanding the individual differences that influence outcome expectancies may enable future interventions to take these factors into account when attempting to reframe OEs (e.g., by simultaneously addressing sexist beliefs). Women's POE and associated traits may be worth further investigation from the perspective of female-on-male harassment.

A breakdown of vignettes individually did not reveal many differences between the sexes, with these being constrained to positive OEs (with moderate effect sizes) associated with relationship disruption. This suggests that it is not any one area of concern, but rather an accumulation of small differences, particularly within positive OEs, which are associated with problematic views and intent. Women appear more likely to perceive persistence from an ex-partner as indicative of harassment. From social and evolutionary psychological perspectives, differences in ratings of relationship disruption behaviours may reflect women's greater suffering of reputation damage when labelled as promiscuous and their greater investment in maintaining a long-term relationship [87, 88].

Persistence is a defining element of sexual harassment within legal guidelines, and distress caused is key to determining where sexual harassment has taken place [89] and has been highlighted as key in women's perceptions of harassment within this study. However, the predominant lack of sex differences in individual vignettes indicates that sex is not the only influential factor. Understanding sexual harassment in the form of persistence and where parameters of persistence differ between men and women may provide insight that could further inform future educational intervention development. A bias toward the saliency of mating opportunities appears related to harassment in men and should be considered within future research. For example, is the influence of high POE a sex difference, or is it indicative of a short-term mating strategy (which are more common, but *not* exclusive to men; [72]). Sociosexuality, having demonstrable links with sexual harassment [72, 90, 91], may be the defining factor influencing the biased impact of POE on H rather than sex itself (although sex is likely a mediatory factor). Thus, this study supports the assessment of interactions between positive and negative outcome expectancies and various individual differences (e.g., sociosexuality to capture the role of sexual strategies) in relation to sexual harassment.

An evolutionary psychological framework of sex differences may contribute to an explanation of this study's results. Both men and women mate in short-term and long-term contexts and have a mating psychology evolved to cope with the demands of each [17, 70]. While this

mating psychology functions similarly in long-term contexts (e.g., both have adaptations for identifying committed mates) it is quite different in short-term contexts. Specifically, ancestral asymmetries in the costs and benefits of casual sex meant that our male ancestors evolved a tendency to seek and capitalise on casual mating opportunities more than our female ones [17, 68]. Consequentially, modern men have inherited biases in perception, decision making, and disposition that may have, historically, increased their ability to secure short-term mates. Men's disposition towards the pursuit of short-term mating helps to explain the marked sex difference in sexual harassment perpetration: because more men than women are interested in pursuing casual sex, men are more likely to be represented among the pool of individuals who harm others in the pursuit of these interests. Understanding the nature of this goal-directed behaviour is important for potential intervention. For example, perspectives which explain sex differences in harassment in terms of power dynamics focus on gender harassment, typically within the workplace [32, 92], and might lead to interventions built under faulty assumptions —that harassment is an assertion of power from those that desire power in and of itself. In contrast, an evolutionary perspective posits the pursuit of casual sex is the key motivating factor that underlies many manifestations of sexual harassment and that striving for power is a tactic to increase one's mate value.

Finally, our vignette measure (the OD-SHAM), while showing good reliability, is newly developed and requires continued psychometric evaluation, perhaps in conjunction with a measure of social desirability [93]. As in previous literature, and as expected given the use of male-on-female harassment vignettes herein, women perceive more behaviours to be higher levels of harassment than men [94]. However, insight into the interplay of OEs and harassment perceptions was gained, and significant sex differences revealed. Men appeared to judge male-on-female sexual interactions based on their belief in the behaviour having a desirable outcome.

## Future directions

Future research should consider alternative perspectives with the OD-SHAM such as reversing character genders to represent female-on-male harassment. Although men are more frequently perpetrators, women also engage in sexual harassment [95], and there is evidence that this may also be somewhat strategic—women's engagement in harassment has been linked to an interest in casual sex, for example [72, 90]. At the same time, women's use of a short-term sexual strategy is qualitatively different to men. Women have casual sex not just for sexual access, but to attract high quality partners for long-term relationships, gain access to protection and resources, and to engage in intrasexual competitions [17, 68] which suggests that a study of the role of OEs in female-on-male sexual harassment may require more nuanced vignettes that factor in these different goals. As discussed, future research should be complemented with further examination of individual differences.

## Limitations

A limitation of our approach is that this cross-sectional design makes it hard to determine causality. It is a reasonable assumption that individuals may base their OEs on whether they perceived the behaviour to be harassment or not. A similar relationship has been demonstrated in the bystander intervention literature whereby the interpretation of an event as problematic seems to precede behavioural decisions [26, 27]. Ultimately, there is likely to be a degree of inter-relatedness between harassment judgements and OEs, though further research could employ longitudinal designs to try to examine this further. It might be the case, for example, that OEs at time 1 are a better predictor of later harassment judgements than the inverse.

Nonetheless, understanding of an individual's OEs, and modification of these, has, as discussed, proven a successful intervention method in a variety of domains [51, 52, 56] and perhaps represents a more efficient means of intervention than challenging harassment perceptions.

## Conclusion

The sex differences found herein indicate the potential of developing interventions that are sex-specific in terms of addressing outcome expectancies (OEs). As positive OEs moderate the impact of negative ones on harassment perception, merely emphasising possible repercussions is unlikely to reduce engagement in harassment. Rather, interventions could adopt a goal-driven approach, providing alternative behaviours that arguably hold a higher positive outcome likelihood. As an example, an intervention tailored to the desire to gain status enhancement significantly reduced bullying behaviours by enabling access to this goal using alternative means [96]. Current health interventions demonstrate that addressing and reducing inaccurate positive OEs reduce problem behaviours [44]. To advance toward intervention development, the generalisability of these effects across contexts (e.g., cross-culturally, reversing sex-roles) and the OEs of those with a history of offending should be examined within future sexual harassment research.

## Supporting information

**S1 Table. Correlations (r) between individual differences and harassment perception and positive and negative outcome expectancies by sex.**
(DOCX)

**S1 Data.**
(SAV)

## Author Contributions

**Conceptualization:** Shonagh Leigh, Andrew G. Thomas, Jason Davies.

**Data curation:** Shonagh Leigh.

**Formal analysis:** Shonagh Leigh, Andrew G. Thomas.

**Funding acquisition:** Andrew G. Thomas, Jason Davies.

**Investigation:** Shonagh Leigh, Andrew G. Thomas, Jason Davies.

**Methodology:** Shonagh Leigh, Andrew G. Thomas, Jason Davies.

**Project administration:** Shonagh Leigh, Andrew G. Thomas, Jason Davies.

**Resources:** Andrew G. Thomas, Jason Davies.

**Supervision:** Andrew G. Thomas, Jason Davies.

**Validation:** Shonagh Leigh, Andrew G. Thomas.

**Visualization:** Shonagh Leigh, Andrew G. Thomas.

**Writing – original draft:** Shonagh Leigh, Andrew G. Thomas.

**Writing – review & editing:** Shonagh Leigh, Andrew G. Thomas, Jason Davies.

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
