## [Decision Letter · Decision Letter 0]

5 Jun 2021

PONE-D-21-09397

The effects of sex and outcome expectancies on perceptions of sexual harassment

PLOS ONE

Dear Dr. White,

Thank you for submitting your manuscript to PLOS ONE. After careful consideration, we feel that it does not fully meet PLOS ONE’s publication criteria as it currently stands. 

A highly-qualified peer reviewer submitted excellent comments on the paper and suggestions for improvement. As academic editor, I reviewed the paper independently. It is clear that this paper is well written and represents a potential contribution to the literature on sexual harassment and positive/negative expectancies, particularly in online dating contexts. At the same time, the reviewer considered the study underdeveloped in terms of the conceptual framework and the literature review on which it is based. The concerns by the reviewer and myself include lack of validation work on the vignette paradigm used, conceptual framework that does not integrate some of the most relevant literature out there, and lack of clarity as to the unique contributions and advancement of knowledge. I do think the focus on positive and negative expectancies borrowed from other literatures (e.g., substance use) and the online dating context could be leveraged in a way that this paper could make an important contribution. For this reason, I would like to extend an opportunity to substantially revise the paper for resubmission, if the various critiques in this letter and in the review can be adequately addressed in a revision. This may be a high bar, as such a revision would involve a reconceptualization of the paper and major re-analysis.

Other comments to address in revision:

The exploratory analyses are under-developed and justified, and seem to be beyond the scope of the conceptual frameworkToo much time spent trying to justify vignette methodology you used and dismissing other methodologies, instead of describing validation of the vignettesInformation is needed on the types of websites and other locations in which the study was advertised: at a university, locally, or anywhere on internet?The various analyses should use a multivariate framework, that include interactions with sex/gender, such as in the hierarchical regression and exploratory analysesThe PxN interaction is decomposed in a way that makes interpretation difficult to translate into real world behaviorsA power analysis is needed to justify the sample size adequacy, especially to detect interactions by sex and PxN interactions (or three way interactions)About 1/3 of your sample identified as non-heterosexual and all your vignettes involved male on female harassment – how does this affect your results?.

In addition, all of the reviewer’s insightful comments should be addressed in the revision and cover letter, if you choose to undertake a revision.

We look forward to receiving your revised manuscript.

Kind regards,

Edelyn Verona

Academic Editor

PLOS ONE

Journal Requirements:

This work was part-funded by the European Social Fund (ESF; https://ec.europa.eu/esf/home.jspcatId=45&langId=en) through the European Union’s Convergence programme administered by the Welsh Government (grant code EGR 0817-100 / EGR9818-100, awarded to SL). The funders had no role in study design, data collection and analysis, decision to publish, or preparation of the manuscript.

Reviewers' comments:

Reviewer's Responses to Questions

**Comments to the Author**

1. Is the manuscript technically sound, and do the data support the conclusions?

Reviewer #1: No

2. Has the statistical analysis been performed appropriately and rigorously? 

Reviewer #1: Yes

3. Have the authors made all data underlying the findings in their manuscript fully available?

Reviewer #1: Yes

4. Is the manuscript presented in an intelligible fashion and written in standard English?

Reviewer #1: Yes

5. Review Comments to the Author

Reviewer #1: The authors use evolutionary psychology as a framework to understand sexual harassment and defining sexual harassment and perceptions of consequences between men and women. Using a quasi-experimental vignette design, they find some overlap between men and women, but also differences with regard to the likelihood of defining behavior as harassment and some of the correlates related to expectations.

There is certainly a voluminous literature that draws on evolutionary psychology to explain sexual violence, but the authors’ rationale and the methodology in this study do not seem to warrant that narrow framework given they are not actually testing motivation from an evolutionary psychology framework. In tailoring their study to an evolutionary psychology framework, they omit much of the relevant research in this area which potentially tempers the contribution of this study. More conceptual development is needed by the authors. Elaboration on the omitted literature and its relevance to the authors study is provided below.

Although the authors use analytic methods that do not assert temporal order, the authors repeatedly state that by including expected outcomes and the perception of a behavior as harassment, that they are providing evidence that individuals define harassment based on the expected positive and negative outcomes of the behavior. However, using this survey design, the authors cannot ascertain temporal order. Why would expectations of outcomes (meeting up or reporting) occur prior to the interpretation of the situation as harassment? It is equally (and perhaps more likely) that individuals interpret the behavior as harassment and that defines their expectations for how the actors will behave. Indeed, drawing on social psychological literature, we would expect that the cognitive order would proceed this way. Ruback and colleagues (1984) theoretical and empirical work on reporting victimization to police outlines a three-stage model where interpretation of the event as problematic is the first step to reporting. This is not unlike the authors dependent variable of reporting harassment. Likewise, the work of Latane and Darley (1969) and the decades of research that has come from that study has asserted that interpreting a situation as problematic is the first step in third-party intervention. Situating their analysis in that framework would necessitate engaging with a body of literature that is not included in this study, but one that is necessary to justify the contribution given their analysis is similar analytically, but not conceptually, to prior work in this area.

The authors attribute the idea that anti-harassment programs do not work because they are not tailored to men. This is not exactly what a lot of the prevention science research on sexual violence finds. Even if tailored, and sometimes especially if tailored, these programs can create a backlash effect. It is not because programs are failing to identify the underpinnings of sexual assault as the authors argue, it is that men do not want to be labeled as perpetrators and when they are subjected to training, they engage in victim blaming as a means of cognitive self-preservation. Dobbin and Kalev have done some excellent work in this area. Likewise, Tinker has also examined backlash as a function of anti-harassment policies. It is unclear how this study specifically relates to sex-specific programming and addresses or refutes the problems found in prior literature.

Likewise, the authors suggest that their study has important implications for programming. Namely, it will provide evidence that sex-specific programs are necessary. One of the conclusions from their analysis is that sex-specific programming may be helpful. Sex-specific approaches are already a best-practice in gender-based violence prevention programming. See below for a few reviews.

Vladutiu CJ, Martin SL, Macy RJ. College-or university-based sexual assault prevention programs: a review of program outcomes, characteristics, and recommendations. Trauma, Violence, & Abuse. 2011;12(2):67–86.

Brecklin LR, Forde DR. A meta-analysis of rape education programs. Violence Vict. 2001;16(3):303–21.

Gibbons R, Evans J. The evaluation of campus-based gender violence prevention programming: what we know about program effectiveness and implications for practitioners. Retrieved from National Online Resource Centre on Violence Against Women’s website: http://www vawnet org. 2013.

Although the majority of the reviewed literature is on workplace harassment (which is the majority of literature on sexual harassment so this is understandable), the authors very briefly mention that they are going to examine harassment in the context of online dating. They include a citation to three studies in that area. This is certainly a potential contribution as research has not thoroughly explored harassment in online contexts. However, there are numerous studies in the area of victimization facilitated by online dating, including perceptions of victimization, reporting, and sex differences. While not always restricted solely to harassment, many of these studies consider a range of behaviors that are considered sexual violence and harassment. The authors do not provide justification for their contribution above and beyond this literature. A few citations are below -

Choi, E. P. H., Wong, J. Y. H., & Fong, D. Y. T. (2018). An emerging risk factor of sexual abuse: the use of smartphone dating applications. Sexual Abuse, 30(4), 343–366.

Clevenger, S. L., Navarro, J. N., & Gilliam, M. (2018). Technology and the endless “cat and mouse” game: A review of the interpersonal cybervictimization literature. Sociology Compass, 12(12), 1–13.

Douglass, C. H., Wright, C. J., Davis, A. C., & Lim, M. S. (2018). Correlates of in-person and technology-facilitated sexual harassment from an online survey among young Australians. Sexual Health, 15(4), 361–365.

Henry, N., & Powell, A. (2018). Technology-facilitated sexual violence: A literature review of empirical research. Trauma, Violence, & Abuse, 19(2), 195–208.

March, E., Grieve, R., Marrington, J., & Jonason, P. K. (2017). Trolling on Tinder®(and other dating apps): Examining the role of the Dark Tetrad and impulsivity. Personality anD Individual Differences, 110, 139-143.

Phan, A., Seigfried-Spellar, K., & Choo, K. K. R. (2021). Threaten me softly: A review of potential dating app risks. Computers in Human Behavior Reports, 3, 100055.

Powell, A., & Henry, N. (2019). Technology-facilitated sexual violence victimization: Results from an online survey of Australian adults. Journal of Interpersonal Violence, 34(17), 3637–3665.

In sum, the authors do not provide adequate justification for this study. I do think they have the potential to make an important contribution if reconceptualized as the data collected seem to be novel. For example, the authors have a wide variety of vignettes that they include in their measure of harassment. Teasing those out to examine how definitions vary across different kinds of behavior and gender-differences in those perceptions would be a study that contributes to the harassment literature and the literature on online victimization, which often is narrowly focused on specific forms of victimization (e.g., cyberstalking). Likewise, the authors include personality scales that are not often considered (but see March et al above), perhaps more development around why those matter and nuanced analyses of those as they relate to harassment and outcomes would inform our current prevention programming.

6. PLOS authors have the option to publish the peer review history of their article (what does this mean?). If published, this will include your full peer review and any attached files.

Reviewer #1: No

---

## [Author Response · Author response to Decision Letter 0]

17 Sep 2021

17/09/2021

Response to reviewers

Thank you for giving us the opportunity to revise our manuscript, titled “The Effects of Sex and Outcome Expectancies on Perceptions of Sexual Harassment”. We have tried our best to address all concerns raised by the reviewers and believe the manuscript has been substantially improved as a result.

COMMENTS TO THE AUTHOR:

Associate Editor:

A highly-qualified peer reviewer submitted excellent comments on the paper and suggestions for improvement. As academic editor, I reviewed the paper independently. It is clear that this paper is well written and represents a potential contribution to the literature on sexual harassment and positive/negative expectancies, particularly in online dating contexts. At the same time, the reviewer considered the study underdeveloped in terms of the conceptual framework and the literature review on which it is based. The concerns by the reviewer and myself include lack of validation work on the vignette paradigm used, conceptual framework that does not integrate some of the most relevant literature out there, and lack of clarity as to the unique contributions and advancement of knowledge. I do think the focus on positive and negative expectancies borrowed from other literatures (e.g., substance use) and the online dating context could be leveraged in a way that this paper could make an important contribution. For this reason, I would like to extend an opportunity to substantially revise the paper for resubmission, if the various critiques in this letter and in the review can be adequately addressed in a revision. This may be a high bar, as such a revision would involve a reconceptualization of the paper and major re-analysis.

Other comments to address in revision:

The exploratory analyses are under-developed and justified, and seem to be beyond the scope of the conceptual framework

As suggested, we have moved away from focusing on an evolutionary psychological concept to a broader framework that accepts sexual harassment is likely multidimensional. We have added more literature to support our examination of positive and negative outcome expectancies (pp5-6). We have also expanded the exploratory analyses to examine individual vignettes and enhanced the discussion around associated traits (p23 for discussion of these). We believe this has greatly improved the overall quality and clarity of the manuscript.

Too much time spent trying to justify vignette methodology you used and dismissing other methodologies, instead of describing validation of the vignettes

We have now included our reliability analysis of the vignette scores (p10) and explained in the introduction why we have chosen a novel vignette measure for our study whilst acknowledging the merit of existing measures (p7).

Information is needed on the types of websites and other locations in which the study was advertised: at a university, locally, or anywhere on internet?

This oversight has been corrected and we now state where our study was advertised in the methods section (p9).

The various analyses should use a multivariate framework, that include interactions with sex/gender, such as in the hierarchical regression and exploratory analyses

Thank you for the suggestion. This is something that we had considered when drafting the paper. However, we felt that examining the sexes separately from the outset was more in line with the nature of the study and the goal of investigating sex differences. There is also precedent for this approach in the sex differences literature as separate analyses of men and women can reveal qualitatively different relationship patterns between variables that simple examination of mean differences often overlooks (Thomas et al., 2021) We now include a better explanation of this throughout the introduction (pp 3-8) and hope that you will deem our reasoning to be satisfactory.

The PxN interaction is decomposed in a way that makes interpretation difficult to translate into real world behaviors

We have added a paragraph in the results explaining the data presented in the simple slopes plot including a worked example which explains what the moderation effect means for responses at the DV level (p20). We agree that these findings, because they are not behavioural, are hard to translate into real world terms However, we hope that our further explanation helps readers get a better feel for the impact of positive outcome expectancies as a moderating force.

A power analysis is needed to justify the sample size adequacy, especially to detect interactions by sex and PxN interactions (or three way interactions)

Thank you. We have now added this to the participants section (p9).We have also added Cohen’s d values to our results section when discussing sex differences found in individual vignettes and aggregate harassment perception scores

About 1/3 of your sample identified as non-heterosexual and all your vignettes involved male on female harassment – how does this affect your results?.

A very good point, one that we checked and have noted in the results section as no qualitative differences were found (p19).

Reviewer #1: 

The authors use evolutionary psychology as a framework to understand sexual harassment and defining sexual harassment and perceptions of consequences between men and women. Using a quasi-experimental vignette design, they find some overlap between men and women, but also differences with regard to the likelihood of defining behavior as harassment and some of the correlates related to expectations.

There is certainly a voluminous literature that draws on evolutionary psychology to explain sexual violence, but the authors’ rationale and the methodology in this study do not seem to warrant that narrow framework given they are not actually testing motivation from an evolutionary psychology framework. In tailoring their study to an evolutionary psychology framework, they omit much of the relevant research in this area which potentially tempers the contribution of this study. 

Thank you for this recommendation. We have revised the introduction to discuss various theories of sexual harassment behaviour, removing the focus on evolutionary psychology. We believe discussing these theories better demonstrates how our research further contributes to this subject (pp 3-8).

More conceptual development is needed by the authors. Elaboration on the omitted literature and its relevance to the authors study is provided below.

Although the authors use analytic methods that do not assert temporal order, the authors repeatedly state that by including expected outcomes and the perception of a behavior as harassment, that they are providing evidence that individuals define harassment based on the expected positive and negative outcomes of the behavior. However, using this survey design, the authors cannot ascertain temporal order. Why would expectations of outcomes (meeting up or reporting) occur prior to the interpretation of the situation as harassment? It is equally (and perhaps more likely) that individuals interpret the behavior as harassment and that defines their expectations for how the actors will behave. Indeed, drawing on social psychological literature, we would expect that the cognitive order would proceed this way. Ruback and colleagues (1984) theoretical and empirical work on reporting victimization to police outlines a three-stage model where interpretation of the event as problematic is the first step to reporting. This is not unlike the authors dependent variable of reporting harassment. Likewise, the work of Latane and Darley (1969) and the decades of research that has come from that study has asserted that interpreting a situation as problematic is the first step in third-party intervention. Situating their analysis in that framework would necessitate engaging with a body of literature that is not included in this study, but one that is necessary to justify the contribution given their analysis is similar analytically, but not conceptually, to prior work in this area.

This is a very good point. We have now included a discussion of temporal order (whilst acknowledging that our study was not able to ascertain this information) and referred to the most likely presentation as outlined in the literature mentioned above (p26). We have also elaborated on our reasoning for the possibility of an alternative temporal order (outcome expectancies influencing harassment judgements). (p26).

The authors attribute the idea that anti-harassment programs do not work because they are not tailored to men. This is not exactly what a lot of the prevention science research on sexual violence finds. Even if tailored, and sometimes especially if tailored, these programs can create a backlash effect. It is not because programs are failing to identify the underpinnings of sexual assault as the authors argue, it is that men do not want to be labeled as perpetrators and when they are subjected to training, they engage in victim blaming as a means of cognitive self-preservation. Dobbin and Kalev have done some excellent work in this area. Likewise, Tinker has also examined backlash as a function of anti-harassment policies. It is unclear how this study specifically relates to sex-specific programming and addresses or refutes the problems found in prior literature.

Thanks to this comment we realise we have not made our intended points clear. We have rephrased and reorganised this section of the introduction with the hope that we have now better explained (pp 3-4). We intended to say that, while interventions are often sex-specific, they are not tailored to men based on their unique sexual psychology, but, as the reviewer says, are tailored to men in the sense that they assume that they are a potential perpetrator. We agree wholeheartedly with the view that this is a counterproductive approach and would like to promote an intervention that first understands those most likely to harass and takes a prosocial goal-orientated approach. It is these goals that we believe may be sex-specific. We have discussed this in relation to some of the literature provided by the reviewer (p 4).

Likewise, the authors suggest that their study has important implications for programming. Namely, it will provide evidence that sex-specific programs are necessary. One of the conclusions from their analysis is that sex-specific programming may be helpful. Sex-specific approaches are already a best-practice in gender-based violence prevention programming. See below for a few reviews.

Vladutiu CJ, Martin SL, Macy RJ. College-or university-based sexual assault prevention programs: a review of program outcomes, characteristics, and recommendations. Trauma, Violence, & Abuse. 2011;12(2):67–86.

Brecklin LR, Forde DR. A meta-analysis of rape education programs. Violence Vict. 2001;16(3):303–21.

Gibbons R, Evans J. The evaluation of campus-based gender violence prevention programming: what we know about program effectiveness and implications for practitioners. Retrieved from National Online Resource Centre on Violence Against Women’s website: http://www vawnet org. 2013.

Although the majority of the reviewed literature is on workplace harassment (which is the majority of literature on sexual harassment so this is understandable), the authors very briefly mention that they are going to examine harassment in the context of online dating. They include a citation to three studies in that area. This is certainly a potential contribution as research has not thoroughly explored harassment in online contexts. However, there are numerous studies in the area of victimization facilitated by online dating, including perceptions of victimization, reporting, and sex differences. While not always restricted solely to harassment, many of these studies consider a range of behaviors that are considered sexual violence and harassment. The authors do not provide justification for their contribution above and beyond this literature. A few citations are below –

Thank you for pointing this out. We agree that there is indeed a rich body of literature on online sexual violence and it was remiss not to mention these within our paper. We have corrected this and clarified that our goal is to take the next steps suggested by these and other papers to focus on that which may inform intervention development (p7). 

Choi, E. P. H., Wong, J. Y. H., & Fong, D. Y. T. (2018). An emerging risk factor of sexual abuse: the use of smartphone dating applications. Sexual Abuse, 30(4), 343–366.

Clevenger, S. L., Navarro, J. N., & Gilliam, M. (2018). Technology and the endless “cat and mouse” game: A review of the interpersonal cybervictimization literature. Sociology Compass, 12(12), 1–13.

Douglass, C. H., Wright, C. J., Davis, A. C., & Lim, M. S. (2018). Correlates of in-person and technology-facilitated sexual harassment from an online survey among young Australians. Sexual Health, 15(4), 361–365.

Henry, N., & Powell, A. (2018). Technology-facilitated sexual violence: A literature review of empirical research. Trauma, Violence, & Abuse, 19(2), 195–208.

March, E., Grieve, R., Marrington, J., & Jonason, P. K. (2017). Trolling on Tinder®(and other dating apps): Examining the role of the Dark Tetrad and impulsivity. Personality anD Individual Differences, 110, 139-143.

Phan, A., Seigfried-Spellar, K., & Choo, K. K. R. (2021). Threaten me softly: A review of potential dating app risks. Computers in Human Behavior Reports, 3, 100055.

Powell, A., & Henry, N. (2019). Technology-facilitated sexual violence victimization: Results from an online survey of Australian adults. Journal of Interpersonal Violence, 34(17), 3637–3665.

In sum, the authors do not provide adequate justification for this study. I do think they have the potential to make an important contribution if reconceptualized as the data collected seem to be novel. For example, the authors have a wide variety of vignettes that they include in their measure of harassment. Teasing those out to examine how definitions vary across different kinds of behavior and gender-differences in those perceptions would be a study that contributes to the harassment literature and the literature on online victimization, which often is narrowly focused on specific forms of victimization (e.g., cyberstalking). 

This is an interesting suggestion and we have now performed analyses and discussed findings of the vignettes individually to give the reader a better feel for the OD-SHAM and to make its components and summary statistics more transparent (pp 11-17).

Likewise, the authors include personality scales that are not often considered (but see March et al above), perhaps more development around why those matter and nuanced analyses of those as they relate to harassment and outcomes would inform our current prevention programming.

We have added to the introduction a discussion of traits associated with sexual harassment in other research and an explanation for the measures used within our study (p8). We also expanded on how revealing associated traits may aid intervention development in the final discussion (p23).

---

## [Decision Letter · Decision Letter 1]

26 Oct 2021

PONE-D-21-09397R1The effects of sex and outcome expectancies on perceptions of sexual harassmentPLOS ONE

Dear Dr. White,

Thank you for submitting your manuscript to PLOS ONE. After careful consideration, we feel that it has merit but does not fully meet PLOS ONE’s publication criteria as it currently stands. Therefore, we invite you to submit a revised version of the manuscript that addresses the points raised during the review process.

Specifically, the same peer reviewer that provided feedback on the last version submitted a new review of the revised manuscript, and I again independently reviewed the revision. As noted by the reviewer, the revisions were highly responsive to comments and provided a balanced overview of the theoretical landscape on harassment research, including highlighting the sex-specific nature of many harassment trainings currently. The revision also provided more information about the development of the vignette paradigm, and other comments were thoroughly addressed. The reviewer had a few more comments during this round of review that I will not repeat here but are worthy to address in another revision before the paper is deemed ready for publication.

One other note, relevant to the reviewer’s suggestion to follow APA or other reference styles guidelines regarding use of language in gender studies, it seems that “gender” rather than “sex” differences are the focus of your study, and thus, this may be another language change to consider: https://apastyle.apa.org/style-grammar-guidelines/bias-free-language/gender.

We look forward to receiving your revised manuscript.

Kind regards,

Edelyn Verona

Academic Editor

PLOS ONE

Journal Requirements:

Reviewers' comments:

Reviewer's Responses to Questions

**Comments to the Author**

1. If the authors have adequately addressed your comments raised in a previous round of review and you feel that this manuscript is now acceptable for publication, you may indicate that here to bypass the “Comments to the Author” section, enter your conflict of interest statement in the “Confidential to Editor” section, and submit your "Accept" recommendation.

Reviewer #1: (No Response)

2. Is the manuscript technically sound, and do the data support the conclusions?

Reviewer #1: Yes

3. Has the statistical analysis been performed appropriately and rigorously? 

Reviewer #1: No

4. Have the authors made all data underlying the findings in their manuscript fully available?

Reviewer #1: Yes

5. Is the manuscript presented in an intelligible fashion and written in standard English?

Reviewer #1: Yes

6. Review Comments to the Author

Reviewer #1: Literature Review

- I found the review of the literature in the revision to be much more balanced. The authors did a commendable job integrating a wider range of theoretical perspectives that research has taken for sexual harassment. The linking aspect of these perspectives of sexual harassment as goal-oriented behavior sets up their study well.

- The rationale for the hypotheses are very brief. The idea that sexual harassment would be associated with outcome expectancy is discussed. The second two ideas – moderation effects and sex differences – need some further justification and elaboration. Likewise, while the authors discuss OEs at length, they do not discuss the importance of defining behaviors as harassment and why linking those two concepts together is important. While it may seem obvious as to why it matters, it does not come across in the manuscript. The discussion of defining behaviors as harassment should also include a sex-specific component given H1 is hypothesizing sex differences in definition that are not discussed in the literature review.

Results

- The comparisons in Table 1 comprise many comparison statistical tests between men and women. It does not appear that the significance of the p-value was adjusted to account for this.

Minor and Miscellaneous

In line with recent guidance from several reference styles (e.g., APA), consider using “men” and “women” as nouns and “males” and “females” as adjectives to avoid biased language. https://apastyle.apa.org/style-grammar-guidelines/bias-free-language/gender

7. PLOS authors have the option to publish the peer review history of their article (what does this mean?). If published, this will include your full peer review and any attached files.

Reviewer #1: No

---

## [Author Response · Author response to Decision Letter 1]

24 Nov 2021

"One other note, relevant to the reviewer’s suggestion to follow APA or other reference styles guidelines regarding use of language in gender studies, it seems that “gender” rather than “sex” differences are the focus of your study, and thus, this may be another language change to consider: https://apastyle.apa.org/style-grammar-guidelines/bias-free-language/gender."

 - This is an important point and we have made the effort to ensure our wording is appropriate throughout. We asked participants for both their assigned biological sex and gender identity, but the analyses groups were formed based on biological sex as opposed to gender. We believe this is fitting with the evolutionary theory applied within the discussion. However, we have clarified this within the text by clearly stating we used “assigned biological sex” for analyses (p.10). We also reran the analyses using gender identity which was not qualitatively different and would not change the overall interpretation of the results. Changes to findings based on gender have now been noted where relevant throughout the results section.

"The rationale for the hypotheses are very brief. The idea that sexual harassment would be associated with outcome expectancy is discussed. The second two ideas – moderation effects and sex differences – need some further justification and elaboration." 

- Thank you for pointing this out, we now realise there was an imbalance between our hypotheses’ rationale. We have expanded on why moderation effects and sex differences are part of our hypotheses and believe this section will be much clearer to the reader as a result (pp.8-9).

"Likewise, while the authors discuss OEs at length, they do not discuss the importance of defining behaviors as harassment and why linking those two concepts together is important. While it may seem obvious as to why it matters, it does not come across in the manuscript."

- This is a good point. As stated by the reviewer, it may seem obvious to us as the writers, but we should not assume the same of all readers. We now have a paragraph discussing this matter in the introduction and believe this makes the overall point of our research much more apparent (p.4).

"The discussion of defining behaviors as harassment should also include a sex-specific component given H1 is hypothesizing sex differences in definition that are not discussed in the literature review."

- As suggested, we have noted sex differences in perceptions of sexual harassment behaviours in this new paragraph.

Results

"The comparisons in Table 1 comprise many comparison statistical tests between men and women. It does not appear that the significance of the p-value was adjusted to account for this."

- We have corrected this oversight throughout all analyses which makes the subsequent findings that much more robust.

Minor and Miscellaneous

"In line with recent guidance from several reference styles (e.g., APA), consider using “men” and “women” as nouns and “males” and “females” as adjectives to avoid biased language. https://apastyle.apa.org/style-grammar-guidelines/bias-free-language/gender"

- Thank you for providing these guidelines. We have made the necessary corrections within the manuscript.

"Please review your reference list to ensure that it is complete and correct. If you have cited papers that have been retracted, please include the rationale for doing so in the manuscript text, or remove these references and replace them with relevant current references. Any changes to the reference list should be mentioned in the rebuttal letter that accompanies your revised manuscript. If you need to cite a retracted article, indicate the article’s retracted status in the References list and also include a citation and full reference for the retraction notice."

- We have checked the references and, whilst none of the papers appear to have retracted, we did find some links that have stopped working since we wrote them and one paper that was a preprint but has now been published. We have updated these accordingly and hope that this addresses the matter.

---

## [Editor Report · Decision Letter 2]

2 Dec 2021

The effects of sex and outcome expectancies on perceptions of sexual harassment

PONE-D-21-09397R2

Dear Dr. White,

We’re pleased to inform you that your manuscript has been judged scientifically suitable for publication and will be formally accepted for publication once it meets all outstanding technical requirements.

Kind regards,

Edelyn Verona

Academic Editor

PLOS ONE
---

## [Editor Report · Acceptance letter]

6 Dec 2021

PONE-D-21-09397R2 

The effects of sex and outcome expectancies on perceptions of sexual harassment 

Dear Dr. Leigh:

I'm pleased to inform you that your manuscript has been deemed suitable for publication in PLOS ONE. Congratulations! Your manuscript is now with our production department. 

Kind regards, 

on behalf of

Dr. Edelyn Verona 

Academic Editor

PLOS ONE